# The Effect of Exercise Intervention on Reducing the Fall Risk in Older Adults: A Meta-Analysis of Randomized Controlled Trials

**DOI:** 10.3390/ijerph182312562

**Published:** 2021-11-29

**Authors:** Mingyu Sun, Leizi Min, Na Xu, Lei Huang, Xuemei Li

**Affiliations:** 1School of Sports Medicine and Rehabilitation, Beijing Sport University, Beijing 100084, China; sunmingyu0531@126.com (M.S.); 18865474685@163.com (N.X.); 2Division of Sport Science & Physical Education, Tsinghua University, Beijing 100084, China; mlz20@mails.tsinghua.edu.cn; 3School of Education and Arts, Minxi Vocational & Technical College, Longyan 364000, China; hl18950808824@126.com

**Keywords:** exercise intervention, fall risk, older adults, meta-analysis

## Abstract

Exercise intervention has a positive effect on reducing the fall risk in older adults. To investigate the effect of different factors of exercise intervention (type, duration, and frequency) on reducing the fall risk in older adults, a meta-analysis was performed in this study. According to the PRISMA^®^, two researchers independently searched PubMed, Web of Science, and the China National Knowledge Infrastructure databases to assess the quality of the studies using the PEDro scale. A total of 648 subjects in 10 randomized controlled trials were included in this study, and the exercise interventions included integrated training (resistance training, core training, and balance training), balance training, core training, Pilates, Ba Duan Jin, and Tai Chi. These studies show that exercise intervention has a huge and significant impact on reducing the risk of falls of the elderly. In conclusion, an integrated intervention with a frequency of more than five times a week and a duration of more than 32 weeks are more effective in reducing the fall risk.

## 1. Introduction

Although falls can occur at all ages, older adults are more prone to injury due to physiological changes and delayed functional recovery caused by aging [1]. Currently, falls are the leading cause of injury and death in people over 65 years old [2]. It has been shown that among older adults aged 65–74 years, 25% experience a fall each year, rising to 29% among those aged 75–84 years and up to 39% in the older adults over 85 years old [3]. Falls in older adults can cause irreversible physical injuries [4], which may even lead to disability or death [5]. The disadvantages of falls affect their quality of life [6] and impose a heavy economic burden on the health care system [4]. In addition, older adults who experience falls also develop a significant fear of falling. Their social isolation and depression can increase the risk of falling, thus creating a vicious cycle [7]. Therefore, the prevention of falls in the elderly is of great significance, and it is urgent to study how to reduce the risk of falls in the elderly. However, the current design of the city is not conducive to the prevention of falls for the elderly to a certain extent, and has a negative impact on the well-being of the elderly, including the lack of access to daily services, and the buildings cannot provide enough space for the elderly and people with mobility impairments [8]. One way to solve the issues above and support the elderly to live a satisfied life is to build elderly-friendly cities and communities [9]. Studies have shown that communities with better street connectivity and daily life services tend to improve physical activities of the elderly [10], and communities roads with low-lying edges and barrier-free sidewalks will improve the travel independence of the elderly [11]. In addition, enhanced intra-city mobility (such as walking ability, use of public transportation) can enhance strengthen the abilities of the elderly in local communities [8]

The World Health Organization has classified fall risk factors into four categories: biological, behavioral, environmental, and socioeconomic factors [12]. The main biological factors of falls include decreased postural balance, sensory processing disorder, muscle weakness, and decreased agility [13]. Currently, the most common interventions that effectively reduce the fall risk are: improving the housing environment to reduce the probability of falls, stopping psychotropic medications, increasing exercise frequencies, and providing supplementation for older adults with vitamin D deficiency [14]. Among these interventions, exercise as a single intervention is considered to be effective and cost-efficient [15]. Many studies have been conducted to explore the effect of exercise interventions on reducing the fall risk in older adults. It has been demonstrated that skeletal muscles are sensitive to mechanical stimuli generated by strength training and are trainable. Older adults with muscle weakness can significantly reverse sarcopenia with strength training [16]. Exercise interventions that target balance, gait, and muscle strength increase can effectively prevent falls in older adults [16]. Among different interventions, the best ones are balance enhancement and lower-extremity resistance training [17]. Additionally, structured exercise interventions such as group exercise under guidance, home training, and Tai Chi can reduce the probability of falls [13]. Dance training, such as Thai traditional dance, can improve balance in older adults [18], and waist functional training can also improve balance and obstacle avoidance ability in older women [19]. A fall prevention program that includes strength and balance training and patient education can improve muscle balance and mental capacity in older women with a history of falls [20]. From a long-term perspective, the effects of exercise interventions on reducing the fall risk in older adults also require consideration of training durations and training frequencies [21].

There are several studies based on older people participating in physical training to reduce the risk of falls, such as: anticipatory control, dynamic stability, functional stability limitations, response control, and flexibility. However, the subject of our research is the risk of falls in the elderly through exercise. Sports include not only physical training, but also Tai Chi, Baduan Jin, yoga, Muay Thai dance and other events. However, our research includes as much exercise as possible [22]. Some scholars reported that compound exercises and Tai Chi are an evaluation of the effect of reducing the risk of falls in the elderly, but they did not mention the effect of the length of exercise intervention and the frequency of exercise on reducing the risk of falls in the elderly. In addition, the outcome indicator chosen in this study is the rate of falls, which is a less subjective. Many falls that did not cause serious injury are often missed by the elderly, so the reported fall rate will be much lower than the actual situation. However, our research chose more scientific outcome indicators, such as: PPA test score, Sensory Organization Test (SOT) test score, 30s chair stand (CS-30) test score, Tetra fall index, Morse Fall Scale (MFS), and fall risk score. These indicators have been proven to be effective objective predictors [23]. Studies have explored the impact of multi-factor interventions on reducing the risk of falls in the elderly. Multi-factor interventions include the following components: exercise, education, environmental modification, medication, walking aids, and vision and psychological management. Sport is only one aspect of it. Since it is difficult to use a multi-factor plan to explain the specific impact of each influencing factor on the results, we prefer to explore the impact of exercise on reducing the risk of falls in the elderly, rather than a multi-factor plan [24]. Compared with this study, there are more elderly people with dementia included in related studies. However, the focus of our discussion is on the elderly who can exercise on their own, excluding those with dementia. In addition, for elderly people with dementia, there are safer ways to reduce their risk of falling. If they are not taken care of by the nursing staff, exercise activities can easily cause them to be injured [25]. Studies have explained which dimensions of abilities of the elderly are improved by exercise, such as dynamic balance, static balance, participants fear of falling, balance confidence, quality of life, and physical performance. However, the study did not indicate which sports the elderly participate in can reduce the risk of falls in the elderly [26].

Although researchers have done a large number of studies investigating the effects of exercise interventions on reducing the fall risk in older adults, few studies have examined the specific effects of different types of exercise interventions. Some studies have pointed out that different forms of exercise have different effects on reducing the risk of falls, but this study only reports the effects of balance, functional exercises and Tai Chi exercises in reducing the risk of falls. It does not reduce the risk of falls for exercises such as resistance training, dancing or walking [16]. Therefore, the aim of this meta-analysis is to investigate the effect of different exercise interventions on reducing the fall risk in older adults.

## 2. Materials and Methods

The systematic evaluation and meta-analysis followed the Preferred Reporting Items for Systematic Evaluation and Meta-Analysis (PRISMA) guidelines and the Cochrane Collaboration protocol. The scheme for this meta-analysis was registered with PROSPERO under the registration number CRD42021279113.

### 2.1. Search Strategy

Two researchers (M.S. and L.M.) independently searched the China National Knowledge Infrastructure (CNKI) database using the Chinese keywords sports, sports activities, physical exercises, fall risk, older adults, middle-aged and older adults. The time frame for the literature search was 1 January 1990 to 31 July 2021. In addition, the researchers searched PubMed and Web of Sciences databases using the keywords “sport”, “physical activity”, “physical fitness”, “fall risk”, “the aged”, “elder population”, “the old”, “middle-aged and senior people”. The time frame was also 1 January 1990 to 31 July 2021.

### 2.2. Inclusion Criteria and Exclusion Criteria

The specific criteria for inclusion and exclusion are shown in Table 1, and the screening process is shown in Figure 1.

### 2.3. Quality Assessment

Two researchers (M.S. and L.M.) independently used the PEDro scale [27] to assess the quality of the included studies. The scale contains 11 items, each item is scored 1 point for meeting the criteria, and the maximum score is 10 (the first item is not scored). Studies were classified as high quality (≥7 points), medium quality (5–6 points), and low quality (≤4 points) based on the scores. When the two researchers disagree, they joined a third party (N.X.), and finally discussed and agreed.

### 2.4. Data Analysis

The data on outcome indicators extracted from the studies were processed using Revmansoftware (version5; https://training.cochrane.org/online-learning/core-software-cochrane-reviews/revman, accessed on 9 January 2021). The outcome variables in this study were continuous. Due to the non-uniformity of unit magnitudes, the standardized mean difference (SMD) was chosen for processing. The specific data were processed according to the following equations:
$$M_{change}=M_{final}-M_{baseline}$$
$$SD_{change}=\sqrt{SD_{baseline}^{2}+SD_{final}^{2}-2\times r\times SD_{baseline}\times SD_{final}}$$
where *M_change_* is the mean of change; *M_baseline_* is the mean before the experiment; *M_final_* is the mean after the experiment; *SD_change_* is the standard deviation of change; *SD_baseline_* is the standard deviation before the experiment; *SD_final_* is the standard deviation after the experiment; r is the correlation coefficient before and after the experiment, which was set to a constant of 0.5 for simplicity of calculation.

The statistic I^2^ was chosen to indicate the heterogeneity among the included studies. I^2^ reflects the degree of overlap of the credible interval. It does not depend on the true effect size and distribution, but reflect the size of the variance on a relative scale [28]. When I^2^ = 0, there was no heterogeneity among studies; when 0 < I^2^ ≤ 20%, there was low heterogeneity among studies; when 20% < I^2^ ≤ 50%, there was medium heterogeneity among studies; when I^2^ > 50%, there was high heterogeneity among studies. When I^2^ > 50%, a random-effects model was chosen, and subgroup analysis and sensitivity analysis were performed to explore the sources of heterogeneity. Funnel plots were constructed for the risk of bias assessment.

## 3. Results

### 3.1. Basic Information of the Included Studies

#### 3.1.1. Characteristics of the Included Studies

The search results were screened and reviewed according to the inclusion and exclusion criteria in this study. A total of 10 studies were finally selected, including four in Chinese and six in English. Participants were 60–95 years old, and different exercise interventions were involved (Table 2).

#### 3.1.2. Characteristics of the Interventions in Included Studies and Outcome Indicators

The duration of the interventions in the included studies ranged from 8 weeks to 1 year. The frequencies of the intervention were mostly three times a week. Outcome indicators included PPA test score, Sensory Organization Test (SOT) test score, 30s chair stand (CS-30) test score, Tetra fall index, Morse Fall Scale (MFS), and fall risk score (Table 3).

### 3.2. The Effect of Exercise Intervention on Reducing the Fall Risk in Older Adults

A combined effect size test was performed on 10 included studies using Revman 5.3 software. The results showed high heterogeneity among studies (I^2^ = 92%, *p* < 0.00001). Therefore, a random-effects model was selected for effect size combining. The results showed that SMD = 1.40, 95% CI interval is (0.75, 2.05), and total effect test Z = 4.22, *p* < 0.0001. Therefore, it can be concluded that exercise intervention can reduce the fall risk in older adults. However, due to the high overall heterogeneity, subgroup analysis was needed to explore the source of heterogeneity (Table 4 and Figure 2).

### 3.3. Publication Bias Assessment

Funnel plots were constructed with the outcome indicators in the included studies, as shown in Figure 3. If the points on the funnel chart are symmetrically scattered around the estimated true value of each independent research effect point, there is an inverted symmetrical funnel shape showing that the included study is unbiased and vice versa. For a symmetrical funnel chart, the greater the degree of bias. The data are largely well symmetrical, indicating that there was some publication bias in the study, but the degree of bias was minimal.

### 3.4. Results of Subgroup Analysis of Effect Sizes

#### 3.4.1. The Effect of Different Types of Exercise Interventions on Reducing the Fall Risk in Older Adults

Exercise interventions can effectively reduce the fall risk in older adults [38], but the value of different types of exercise activities differs for older adults [39]. Therefore, it was assumed in this study that differences in the type of exercise interventions could have different effects on reducing the fall risk in older adults. According to the different types of exercise interventions, we divided them into three subgroups: integrated training (resistance training, core training, and balance training), physical training and fitness training (Pilates, Ba Duan Jin, and Tai Chi). The subgroup analysis results showed that the within-group heterogeneity was high for integrated training (I^2^ = 95%) and physical training (I^2^ = 85%), and it was low for fitness training (I^2^ = 31%), indicating that all kinds of exercise interventions were effective in reducing the fall risk in older adults. In terms of the SMD, integrated training (SMD = 3.16) > physical training (SMD = 0.88) > fitness training (SMD = 0.57), as shown in Table 5 and Figure 4.

#### 3.4.2. Effect of Different Intervention Durations on Reducing the Fall Risk in Older Adults

From a long-term perspective, the effectiveness of exercise interventions to reduce the fall risk in older adults is related to the duration of the intervention [21]. The included studies were analyzed in subgroups according to different durations of interventions. The results showed that there was a high within-group heterogeneity for intervention durations of less than 12 weeks (I^2^ = 77%), 12–32 weeks (I^2^ = 88%), and more than 32 weeks (I^2^ = 96%), indicating that different intervention durations have a significant effect on reducing the fall risk in older adults. In terms of the SMD, intervention durations more than 32 weeks (SMD = 2.92) > 12–32 weeks (SMD = 0.98) > less than 12 weeks (SMD = 0.68), as shown in Table 6 and Figure 5.

#### 3.4.3. Effect of Different Intervention Frequencies on Reducing the Fall Risk in Older Adults

Similarly, the effects of exercise interventions on reducing the fall risk in older adults are related to the frequency of the intervention [21]. By subgroup analysis of the included studies according to different exercise intervention frequencies, it was shown that a frequency of 3–5 times a week (I^2^ = 85%) and a frequency of more than 5 times a week (I^2^ = 96%) had high within-group heterogeneities. This result indicated that different intervention frequencies have significant effects on reducing the fall risk in older adults. In terms of the SMD, more than five times a week (SMD = 2.39) > 3–5 times a week (SMD = 1.17), as shown in Table 7 and Figure 6.

## 4. Discussion

Normal human aging is associated with the decline of many physiological functions, including the decline of skeletal muscles, cardiovascular, vision, vestibular system, and proprioception. These declines result in decreased body coordination, slower postural responses, and cognitive decline, which are associated with increased fall risk [40]. Therefore, a comprehensive fall prevention program is essential for older adults [41]. Currently, assessing the fall risk of older adults is a valuable tool for caregivers to take targeted and effective measures [42]. However, in different situations, researchers tend to choose different fall risk assessment tools [2]. For example, the TUG test has been widely adopted by researchers due to its lower cost, simplicity of the procedure, and excellent sensitivity [43]. Besides, gait balance evaluation based on MoS [44] and Tinetti test [45] are more suitable for individual balance assessment. The fall risk assessment in some studies has also included both physical and perceived dimensions [46]. In the included studies, a wide variety of test metrics were involved, and the scales were not uniform. It can be speculated that the different fall risk assessments are a source of heterogeneity between studies, which needs to be verified in future studies.

This study shows that integrated training (resistance training, core training, and balance training) is more effective in reducing the fall risk in older adults, consistent with previous studies [13]. Previous studies have also shown that adherence rates of participants may be higher in exercises with shorter durations [47], while the present study shows that exercise interventions are more effective in reducing the fall risk in older adults when the duration of training is 32 weeks or more. Considering whether exercise interventions produce a conscious continuation of exercise behavior, future studies should focus on the relationship between the effectiveness and the cost-effectiveness of exercise interventions on reducing the fall risk in older adults.

Previous studies have demonstrated that the effect of exercise interventions in reducing the fall risk in older adults is closely related to the exercise load [16]. Few studies have shown that exercises with high loads produce better results [48]. In this study, most of the selected studies addressed exercise duration and frequency, and only a few included studies were related to the intervention load. Therefore, the intervention load should be considered in future studies.

The research is more for relatively healthy elderly people. Our results are not applicable to elderly people who are dementia, stroke rehabilitation, disabled or other people who cannot exercise independently. Future studies can be done to explore more rehabilitation options for elderly people who cannot exercise independently. In addtion, the study lacks more detailed subgroup analysis such as the duration of each training and the chronic disease of the trainer due to the limited number of included studies.

In conclusion, reducing the fall risk in older adults should be aimed at reducing and eliminating fall risk factors [49]. Since the causes of falls in older adults are usually the result of multiple factors, exercise as a single intervention has reduced the incidence of falls by more than 36% [25], some studies have pointed out that multifactorial interventions are more effective than single-factor interventions [50]. Interventions such as dual-task training [51], multiple standardized prevention programs (including exercise components and individualized design) [13], and multifactorial fall prevention programs with exercise interventions [52] are beneficial for older adults. Factors can include sports, education, environmental transformation, drug treatment, walking aids, vision and psychological management, etc. [24]. Therefore, multiple factors should also be considered to explore the effect of exercise interventions on reducing the fall risk in older adults.

## 5. Conclusions

The exercise intervention is effective in reducing the fall risk in older adults. Integrated training (resistance training, core training, and balance training) with a duration of over 32 weeks and a frequency of more than five times a week is more effective in reducing the fall risk in older adults. It is a long-term and continuous method to reduce the risk of falls of the elderly through exercise. Researchers need to consider economic conditions and the willingness of the elderly when choosing exercise methods for intervention.

## Figures and Tables

**Figure 1 ijerph-18-12562-f001:**
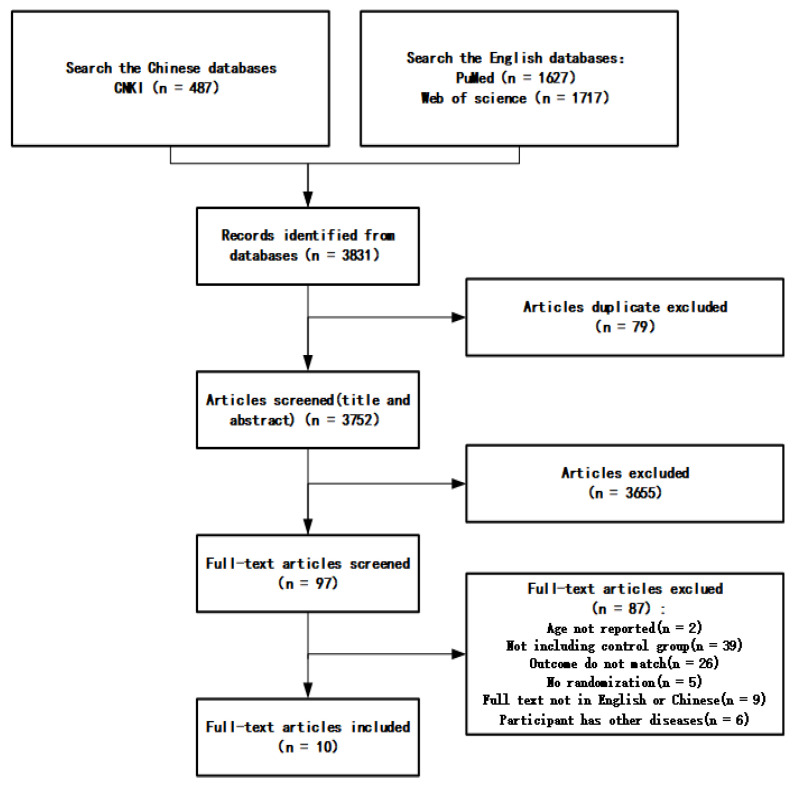
Screening process.

**Figure 2 ijerph-18-12562-f002:**
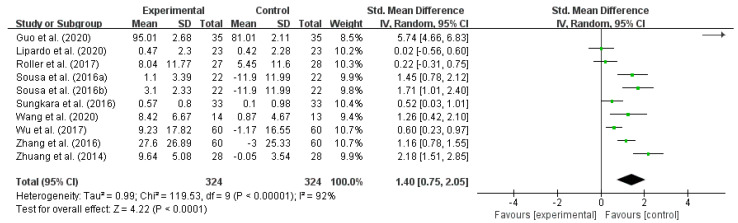
The effect of exercise intervention on reducing the fall risk in older adults. Notes: The figure is automatically generated, the bold no meaning.

**Figure 3 ijerph-18-12562-f003:**
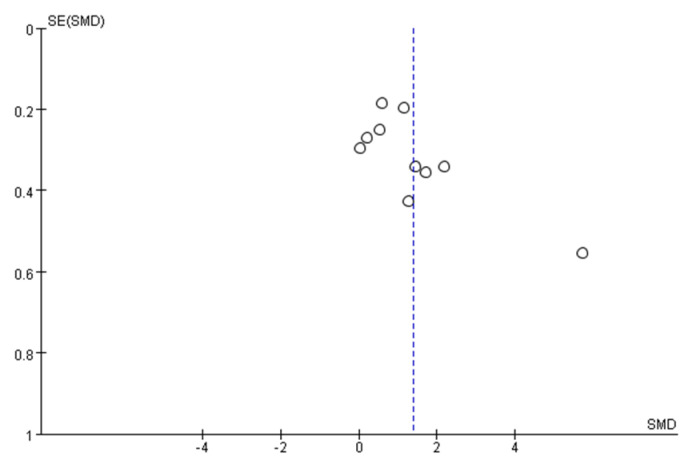
Funnel plot of publication bias assessment.

**Figure 4 ijerph-18-12562-f004:**
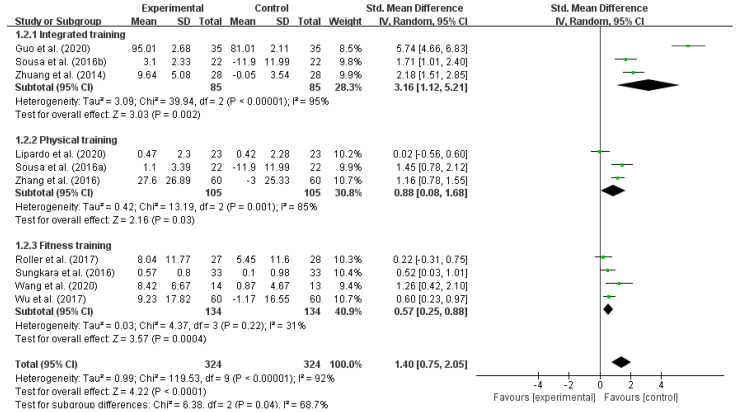
Effect of different exercise interventions on reducing the fall risk in older adults. Notes: The figure is automatically generated, the bold no meaning.

**Figure 5 ijerph-18-12562-f005:**
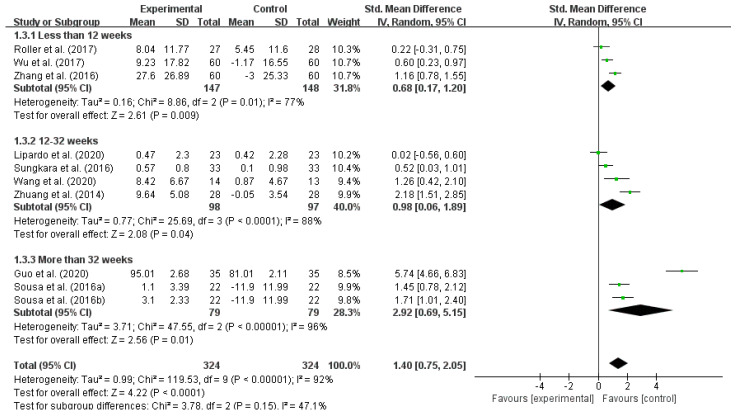
Effect of different intervention durations on reducing the fall risk in older adults. Notes: The figure is automatically generated, the bold no meaning.

**Figure 6 ijerph-18-12562-f006:**
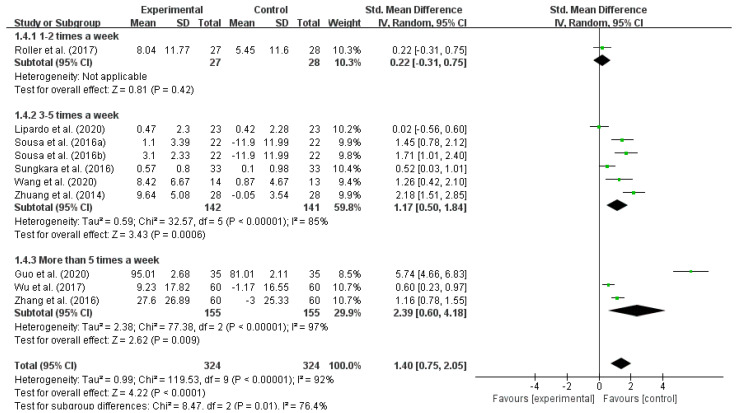
Effect of different intervention frequencies on reducing the fall risk in older adults. Notes: The figure is automatically generated, the bold no meaning.

**Table 1 ijerph-18-12562-t001:** Inclusion criteria and exclusion criteria.

Criteria	Type	Definition
Inclusion Criteria	Study type	Randomized controlled trial (RCT) studies in Chinese and English.
	Study subject	Middle-aged and older adults (over 50 years old) without disabilities and other diseases that make them unsuitable for exercise interventions.
	Intervention requirement	Physical activity at baseline levels is the same in both groups, and the duration of the intervention is greater than or equal to 4 weeks.
	Outcome indicators	Fall risk index, fall risk score.
Exclusion Criteria		Indicators without clear comparability.
		Baseline level data not available.
		Missing full text or incomplete data on outcome indicators.

**Table 2 ijerph-18-12562-t002:** Characteristics of the included studies.

Code	Publish Time	Country	intervention Programs	Exercise Intervention Type	Participant Type	Age	Gender	Test Group Size	Control Group Size	PEDro Score	Literature Quality
Guo et al. (2020) [29]	2020	China	Exercise includes weight-bearing exercise, balance training and muscle training. Weight-bearing exercises are carried out 3 times a day, 20 min each time. Balance exercise can improve balance ability, which is carried out 3 times a day, 20 min each time.	Integrated training (weight training, balance training, and muscle training)	Older adults with osteoporosis	62–80	Mixed	35	35	5	Medium
Lipardo et al. (2020) [30]	2020	Philippines	12 consecutive weeks, 3 sessions per week, 60 to 90 min of physical training	Physical training	Older adults in the community with mild cognitive impairment	60–83	Mixed	23	23	5	Medium
Roller et al. (2017) [31]	2018	United States	Subjects in the Pilates group attended 8 × 10³sessions of a 45-min Pilates exercise program using a Reformer once a week over a 10-week period	Pilates	Older adults over 65 years old with a history of falls in the past year	65–95	Mixed	27	28	7	High
Sousa et al. (2016a) [32]	2016	Portugal	The aerobic exercise group trained twice per week in a land environment (Mondays and Wednesdays) and once per week in an aquatic environment (Fridays). All aerobic training sessions consisted of: (i) a 10-min warm-up period, which included walking and flexibility exercises; (ii)a 30-min cardiorespiratory period, including walking and/or brisk walking; (iii) a 10-min muscular endurance, which included three exercises (three sets of 15–20 repetitions) using only bodyweight and gravity for strengthening the lower and upper limbs in a land environment, and water resistance in an aquatic environment; and (iv) a 5-min cool-down period, which included breathing and stretching exercises.	Aerobic exercise	Older men in the community	65–74	Male	22	22	6	Medium
Sousa et al. (2016b) [32]	2016	Portugal	the aerobic training session on Mondays were replaced by a resistance exercise session. The intensity of the resistance training sessions was defined by the pyramidal method set to 65% of 1-RM in the first eight weeks; 75% of 1-RM for Weeks 8–24; 70% of 1-RM for Weeks 24–28; and 65% of 1-RM for Weeks 28–32 (three sets of 10–12 repetitions in all sessions). Each session always began with a 10-min warm-up and ended with a cool-down period. The main part of the sessions consisted of a circuit of seven exercises: bench press, leg press, lateral pull-down, leg extension, military press, leg curl and arm curl, in this order, and carried out with conventional variable resistance devices (PANATTA, Fitline 2000 series, Italy). The 1-RM values were taken at the first workout of Week 1, Week 8, Week 16 and Week 24, and at the last workout of Week 32, allowing periodic adjustment of the resistance training intensity for the combined exercise group.	Integrated training (weight training, balance training, and muscle training)	Older men in the community	65–74	Male	22	22	6	Medium
Sungkarat et al. (2016) [33]	2017	Thailand	Participants in the Tai Chi group attended Tai Chi classes led by a certified Tai Chi instructor for 3 weeks (9 sessions) to learn Tai Chi principles and the 10-form Tai Chi. The Tai Chi classes were held with six or seven persons per class in the exercise room at the Department of Physical Therapy. Participants then practiced Tai Chi at home three times per week for 12 weeks.	Tai chi	Older adults with mild cognitive impairment	61–75	Mixed	33	33	6	Medium
Wang et al. (2020) [34]	2020	China	Exercise 5 times a week, time is 7:00–8:30 in the morning, each exercise time is 90 min, including 15 min of warm-up activities, 60 min of formal exercises, and 15 min of relaxation.	Diabolo training	Older adults in rural area	60–72	Female	14	13	5	Medium
Wu et al. (2017) [35]	2017	China	Baduanjin exercises are performed 2 times a day, 3 times each time, and the treatment cycle is 30 days. The patient chooses appropriate actions for practice under the guidance of doctors according to their own conditions and individual differences	Ba Duan Jin	Older adults in the community	65–80	Mixed	60	60	6	Medium
Zhang et al. (2016) [36]	2016	China	Training time is 45 min, training 5 days per week, and the intervention time is 8 weeks	Balance training	Older adults in the community	63–78	Mixed	60	60	6	Medium
Zhuang et al. (2014) [37]	2014	China	The intervention group received 60 min exercise classes three times per week for 12 weeks in the community center. The intensity of the exercise was increased gradually over time to help participants achieve success. Each class started with a 5-min warm-up, followed by 15 min of balance exercises, 15 min of muscle-strength training, 15 min of 8-form Yang style Tai Chi Chuan, and ending with 10 min of flexibility/stretching and cool-down.	Integrated training (muscle training, balance raining, and Tai Chi)	Older adults in the community	60–80	Mixed	28	28	6	Medium

**Table 3 ijerph-18-12562-t003:** Characteristics of interventions and outcome indicators in the included studies.

Code	Main Indicator	Outcome Indicator	Intervention Duration	Intervention Frequency	Result
Guo et al. (2020) [29]	Comparison of disease efficacy, disease awareness, fall prevention awareness, psychological anxiety level, body immunity status, bone density before and after care, and incidence of falls between the two groups of older adults with osteoporosis.	Fall prevention awareness score	1 year	Daily	Medication and exercise interventions have a better preventive effect on older adults.
Lipardo et al. (2020) [30]	Assessment of fall incidence, overall fall risk, dynamic balance, walking speed, and lower-extremity strength.	PPA (Physiological Profile Assessment) test	12 weeks	3 times a week	No significant differences in fall rates and fall risk were found between the groups after the intervention. Physical training and cognitive training and the combination of them can improve dynamic balance. Further studies with larger sample sizes are needed to determine if the interventions are effective.
Roller et al. (2017) [31]	SOT score of the NeuroCom^®^ system, Timed Up and Go (TUG), and Activity Specific Balance Confidence (ABC) Scale.	SOT score	10 weeks	Once a week	Pilates modified exercises once a week for 10 weeks can reduce the fall risk in adults aged 65 and older who are at risk of falls. This exercise significantly improved static and dynamic balance, functional mobility, balance self-efficacy, and lower-extremity AROM. In contrast, the control group showed no significant improvement in any of the indicators. Pilates modified exercises were more effective than no exercise intervention in improving AROM in the hip and ankle joints.
Sousa et al. (2016a) [32]	Timed up test, functional reach (FR) test, CS-30 test, and 6-min walk test.	CS-30 test	32 weeks	3 times a week	The addition of resistance exercise to aerobic exercise may improve factors associated with increased fall risk. Both combined exercise and aerobic exercise are more effective than no exercise in reducing the fall risk.
Sousa et al. (2016b) [32]	Timed up test, FR test, CS-30 test, and 6-min walk test.	CS-30 test	32 weeks	3 times a week	The addition of resistance exercise to aerobic exercise may improve factors associated with increased fall risk. Both combined exercise and aerobic exercise are more effective than no exercise in reducing the fall risk.
Sungkarat et al. (2016) [33]	Fall risk index with physiological profile assessment (PPA).	PPA test	12 weeks	3 times a week	In older adults with multi-domain amnestic mild cognitive impairment, the combination of core training and Tai Chi three times a week for 15 weeks significantly improved cognitive function and modestly reduced the fall risk. In particular, Tai Chi may have quite great effects on reducing the fall risk.
Wang et al. (2020) [34]	Static and dynamic balance ability test (Good Balance balance tester), CS-30 test, Tetrax fall risk test.	Tetrax fall risk index	14 weeks	5 times a week	The 14-week diabolo exercise strengthened the lower-extremity muscles of older adults, reducing the possibility of the fall risk and improving balance. Eight weeks of physical education and health lectures motivated older adults to take action, increasing their exercise frequencies.
Wu et al. (2017) [35]	MFS, Berg balance scale score, “get-up-and-walk” timing test, self-rating anxiety scale, Riker SAS score, and the SF-36 health survey.	MFS	30 days	Daily	Ba Duan Jin could effectively reduce the fall risk among older adults in the community to a certain extent.
Zhang et al. (2016) [36]	Fall risk score	Fall risk score	8 weeks	Daily	Balance training could reduce the fall risk in older adults.
Zhuang et al. (2014) [37]	CS-30 test, TUG test, FR test, and star offset balance test (SEBT).	CS-30 test	12 weeks	3 times a week	This study provided an effective, evidence-based fall prevention program that can be implemented to improve the physical health of older adults in the community and reduce the fall risk. In the assessment of fall risk in older adults, the SEBT might be a sensitive measure of physical performance.

**Table 4 ijerph-18-12562-t004:** Test for overall combined effect size.

Model	Sample Size	Heterogeneity Test		Total Effect Size Test		SMD	95% CI
		I^2^	*p*	Z	*p*		
Random-Effect Model	10	92%	<0.00001	4.22	<0.0001	1.4	0.75–2.05

I^2^: I squared statistic; Z: overall effect; SMD: standardized mean difference.

**Table 5 ijerph-18-12562-t005:** Effect of different exercise interventions on reducing the fall risk in older adults.

Type	No. of Studies	Test Group Size	Control Group Size	I^2^	SMD	95% CI	*p*
Integrated training	3	85	85	95%	3.16	1.12–5.21	0.002
Physical training	3	105	105	85%	0.88	0.08–1.68	0.03
Fitness training	4	134	134	31%	0.57	0.25–0.88	0.004
Total	10	324	324	92%	1.4	0.75–2.05	<0.0001

**Table 6 ijerph-18-12562-t006:** Effect of different intervention durations on reducing the fall risk in older adults.

Duration	No. of Studies	Test Group Size	Control Group Size	I^2^	SMD	95% CI	*p*
Less than 12 weeks	3	147	148	77%	0.68	0.17–1.20	0.009
12–32 weeks	4	98	97	88%	0.98	0.06–1.89	0.04
More than 32 weeks	4	79	79	96%	2.92	0.69–5.15	0.01
Total	10	324	324	92%	1.4	0.75–2.05	<0.0001

**Table 7 ijerph-18-12562-t007:** Effect of different intervention frequencies on reducing the fall risk in older adults.

Frequency	No. of Studies	Test Group Size	Control Group Size	I^2^	SMD	95% CI	*p*
1–2 times a week	1	27	27	-	0.22	−0.31–0.75	0.42
3–5 times a week	6	142	141	85%	1.75	0.50–1.84	0.0006
More than 5 times a week	3	155	155	97%	2.39	0.60–4.18	0.009
Total	10	324	324	92%	1.4	0.75–2.05	<0.0001

## Data Availability

Not applicable.

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
