# Peer review of "The Effect of Exercise Intervention on Reducing the Fall Risk in Older Adults: A Meta-Analysis of Randomized Controlled Trials"

_ijerph, 2021, doi:10.3390/ijerph182312562_

Round 1

Reviewer 1 Report

For what concerns the style and writing of the manuscript, I have no major concerns. They are overall adequate.

Now, moving on with the contents. In general, I found this manuscript very interesting and following a rigorous research structure. A few comments:

Introduction: It has a good flow. I would suggest that you include some mentions to the detrimental factor cities could be for ageing and wellbeing (for instance, https://doi.org/10.1177/0042098019852033)

Abstract: although very informative, I would like to suggest avoiding statistical formulas in this section. I would rather suggest including what you found in words, for instance, “significant evidence was found for…”

2.1 Search Strategy: is there a reason for the keywords searched in CNKI being different from those searched in PubMed and WOS?

Figure 1: please improve the quality of the graph.

2.4 Data analysis: please mention what is the statistic I2 (not all your readers will be experts in the topic). State that you used the combined effect size test, and, if possible, include some lines to explain it (for instance, DOI: 10.1037/1082-989x.7.1.105 ).

3.1.2 Characteristics of the Interventions in Included Studies and Outcome Indicators: what is PPA test Score? (I suppose it is Personality Profile Assessment test).

Table 3: this table is unreadable.

Line 163: resistance I believe.

Conclusions: do not you believe that training programs over 32 weeks and a frequency of more than 5 times, are very rare and very difficult to follow? Not only in economic terms, but also in what concerns personal motivation to follow the program. Finally, please include a limitation section.

Author Response

Dear reviewer,

Introduction: It has a good flow. I would suggest that you include some mentions to the detrimental factor cities could be for ageing and wellbeing (for instance, https://doi.org/10.1177/0042098019852033)

Reply: Thank you for your suggestions. It is mentioned in the introduction that "it is recommended that you mention some of the factors that may be detrimental to aging and well-being in some cities. The following content is added to the article: "The vast majority of the elderly want to spend the rest of their lives in comfortable and convenient circumstances, which is not fully guaranteed by the current city design. For example, the ongoing urban planning is poorly designed to protect the elderly from unexpected falls. The lack of access to daily services and rows of buildings, which greatly limit the space for the elderly especially those and people with mobility impairments, reflect the negative impacts of the existing city design on old people[24]. One way to solve the issues above and support the elderly to live a satisfied life is to build elderly-friendly cities and communities[25]. Studies have shown that communities with better street connectivity and daily life services tend to improve physical activities of the elderly [26], and communities roads with low-lying edges and barrier-free sidewalks will improve the travel independence of the elderly [27]. In addition, enhanced intra-city mobility (such as walking ability, use of public transportation) can enhance strengthen the abilities of the elderly in local communities [24].".

Abstract: although very informative, I would like to suggest avoiding statistical formulas in this section. I would rather suggest including what you found in words, for instance, “significant evidence was found for…”

Reply: Thank you for raising these questions. Here, “The results showed standardized mean difference (SMD) = 1.40, 95% CI = [0.75, 2.05], total effect test Z = 4.22, p < 0.00001, and I2 = 92%", has been changed into “These studies show that exercise intervention has a huge and significant impact on reducing the risk of falls of the elderly”.

2.1 Search Strategy: is there a reason for the keywords searched in CNKI being different from those searched in PubMed and WOS?

Reply: Thank you for raising this interesting point. The CNKI search keywords mentioned by reviewers are different from PubMed and WOS search keywords. This is due to language difference as CNKI is a Chinese database.

Figure 1: please improve the quality of the graph.

Reply: Thank you for your suggestions. According to the reviewers’ comments, higher resolution pictures are provided below as follows:

2.4 Data analysis: please mention what is the statistic I2 (not all your readers will be experts in the topic). State that you used the combined effect size test, and, if possible, include some lines to explain it (for instance, DOI: 10.1037/1082-989x.7.1.105 ).

Reply: Thank you for raising these questions. We add the following content: I2 reflects the degree of overlap of the credible interval. It does not depend on the true effect size and distribution but reflect the size of the variance on a relative scale.

3.1.2 Characteristics of the Interventions in Included Studies and Outcome Indicators: what is PPA test Score? (I suppose it is Personality Profile Assessment test).

Reply: Thank you for raising this interesting point. An explanation of the PPA test scores has been added to the text, as follows:“which is “Physiological Profile Assessment””.

Table 3: this table is unreadable.

Reply: Thank you for raising these questions. The format of Table 3 has been adjusted.

Line 163: resistance I believe.

Reply: Thank you for raising this interesting point. The question raised in Line 163 can be answered that here is indeed "sports intervention".

Conclusions: do not you believe that training programs over 32 weeks and a frequency of more than 5 times, are very rare and very difficult to follow? Not only in economic terms, but also in what concerns personal motivation to follow the program.

Reply: Thank you for pointing this out. Regarding the conclusion, the explanation was added: "It is a long-term and continuous method to reduce the risk of falls of the elderly through exercise. Researchers need to consider economic conditions and the willingness of the elderly when choosing exercise methods for intervention."

Finally, please include a limitation section.

Reply: Thank you for your suggestions.The limitation part was added: "The study lacks  more detailed subgroup analysis such as the duration of each training, the chronic disease of the trainer, etc. In addition, because of the scarce of samples but complexity of exercise intervention methods, the classification in the subgroup analysis is relatively rough and further researches are needed to explore the contents and effects of comprehensive training."

We thank you for considering this work and look forward to your response.

Sincerely

Reviewer 2 Report

Thank you for the opportunity to read this manuscript. It is a systematic review with the objective to investigate the effect of different exercise interventions on reducing the fall risk in older adults. As the authors say, there is a lot of literature on this but the authors focus on the type of exercise, but I feel that this is a review that does not provide any new information and also has many shortcomings as there is a lack of a more comprehensive review of the literature where the authors would have found this previously published and recent information.

For example, several systematic reviews that address the topic are listed here. If the authors feel that they provide novel information, they should at least review these and other recently published articles and emphasize what the new review provides.

https://pubmed.ncbi.nlm.nih.gov/33186739/

https://pubmed.ncbi.nlm.nih.gov/33239019/

https://pubmed.ncbi.nlm.nih.gov/32272282/

https://pubmed.ncbi.nlm.nih.gov/31982358/

https://pubmed.ncbi.nlm.nih.gov/31792067/

https://pubmed.ncbi.nlm.nih.gov/32796528/

Methods,

I feel that the authors have not performed a correct search equation since very few RCTs were included in the period covered by the search.

Results

Table 3 is not readable

Please check references  

Author Response

Dear reviewer,

For example, several systematic reviews that address the topic are listed here. If the authors feel that they provide novel information, they should at least review these and other recently published articles and emphasize what the new review provides.

https://pubmed.ncbi.nlm.nih.gov/33186739/

https://pubmed.ncbi.nlm.nih.gov/33239019/

https://pubmed.ncbi.nlm.nih.gov/32272282/

https://pubmed.ncbi.nlm.nih.gov/31982358/

https://pubmed.ncbi.nlm.nih.gov/31792067/

https://pubmed.ncbi.nlm.nih.gov/32796528/

Reply: Thank you for pointing this out. We reviewed the literature provided by the reviewer, and fill the following content into the article:

(1) "Studies have proved that compared with the control group, patients receiving physical exercise have improved dynamic balance, static balance, fear of falling, confidence in balance, quality of life, and physical performance [28]."

(2) "Some studies have pointed out that different forms of exercise have different effects on reducing the risk of falls, but this study only reports the effects of balance, functional exercises and Tai Chi exercises in reducing the risk of falls. It does not reduce the risk of falls for exercises such as resistance training, dancing or walking[18]."

(3) "Exercise as a single intervention has reduced the incidence of falls by more than 36%[52],"

(4) Sports, education, environmental modification, medication, walking aids, vision and psychological management, etc.[57]

I feel that the authors have not performed a correct search equation since very few RCTs were included in the period covered by the search.

Reply: Thank you for pointing this out.I checked the search equation again and found that RCTs are indeed very few. Here are the keywords used in literature search : Two researchers (M.S. and L.M.) independently searched the China National Knowledge Infrastructure (CNKI) database using the Chinese keywords sports, sports activities, physical exercises, fall risk, older adults, middle-aged and older adults. The time frame range for the literatures was January 1, 1990 to July 31, 2021. In addition, the researchers searched PubMed and Web of Sciences databases using the keywords “sport”, “physical activity”, “physical fitness”, “fall risk”, “the aged”, “elder population”, “the old”, “middle-aged and senior people”. The time frame range was also January 1, 1990 to July 31, 2021.

Table 3 is not readable

Reply: Thank you for raising these questions. The format of Table 3 has been adjusted.

Please check references  

Reply: Thank you for pointing this out.We have checked the references

Reviewer: 3

In figure 1, the authors could remove the last box.

Reply: Thank you for your suggestions.We removed the last box in figure.

Page 4. it would be interesting if you made a detailed description of the intervention programs.

Table 2. You could change the study code to the name of the first author (e.g. Author et al. (2020)). Thus, the reader can more easily identify the study.

Reply: Thank you for raising this interesting point. We changed the research code in Table 2 to the name of the first author, and add a specific intervention plan.( See the table2 in article)

Table 3 is unreadable. Please could you fix it?

Reply: Thank you for raising these questions. The format of Table 3 has been adjusted.

Page 9. Could you add the forest plots?

Reply: Thank you. We add the following forest plot:

Figure 2. The Effect of Exercise Intervention on Reducing the Fall Risk in Older Adults

Page 9. Could you report some statistics from the funnel plots? In this way, we could know more precisely the publication bias.Page 10.

Reply: Thank you for pointing this out. We added the following content to the bias analysis part: If the points on the funnel chart are symmetrically scattered around the estimated true value of each independent research effect point, there is an inverted symmetrical funnel shape showing that the included study is unbiased and vice versa. For a symmetrical funnel chart, the greater the degree of bias.

Could you report the forest plots for subgroup analyzes?

Reply: Thank you. We add the following forest plots:

Figure 4. Effect of Different Exercise Interventions on Reducing the Fall Risk in Older Adults

Figure 5. Effect of Different Intervention Durations on Reducing the Fall Risk in Older Adults

Figure 6. Effect of Different Intervention Frequencies on Reducing the Fall Risk in Older Adults

We thank you for considering this work and look forward to your response.

Sincerely

Reviewer 3 Report

The purpose of this study was is to investigate the effect of different exercise interventions on reducing the fall risk in older adults. The data collected in this meta-analysis may affirm or expand on available literature. However, there are still few things that need to be corrected.

The authors make a good theoretical foundation and use an appropriate methodology.
In figure 1, the authors could remove the last box.

Results
Page 4. it would be interesting if you made a detailed description of the intervention programs.
Table 2. You could change the study code to the name of the first author (e.g. Author et al. (2020)). Thus, the reader can more easily identify the study.
Table 3 is unreadable. Please could you fix it?
Page 9. Could you add the forest plots?
Page 9. Could you report some statistics from the funnel plots? In this way, we could know more precisely the publication bias.
Page 10. Could you report the forest plots for subgroup analyzes?

Author Response

Dear reviewer,

In figure 1, the authors could remove the last box.

Reply: Thank you for your suggestions.We removed the last box in figure.

Page 4. it would be interesting if you made a detailed description of the intervention programs.

Table 2. You could change the study code to the name of the first author (e.g. Author et al. (2020)). Thus, the reader can more easily identify the study.

Reply: Thank you for raising this interesting point. We changed the research code in Table 2 to the name of the first author, and add a specific intervention plan.( See the table2 in article)

Table 3 is unreadable. Please could you fix it?

Reply: Thank you for raising these questions. The format of Table 3 has been adjusted.

Page 9. Could you add the forest plots?

Reply: Thank you. We add the following forest plot:

Figure 2. The Effect of Exercise Intervention on Reducing the Fall Risk in Older Adults

Page 9. Could you report some statistics from the funnel plots? In this way, we could know more precisely the publication bias.Page 10.

Reply: Thank you for pointing this out. We added the following content to the bias analysis part: If the points on the funnel chart are symmetrically scattered around the estimated true value of each independent research effect point, there is an inverted symmetrical funnel shape showing that the included study is unbiased and vice versa. For a symmetrical funnel chart, the greater the degree of bias.

Could you report the forest plots for subgroup analyzes?

Reply: Thank you. We add the following forest plots:

Figure 4. Effect of Different Exercise Interventions on Reducing the Fall Risk in Older Adults

Figure 5. Effect of Different Intervention Durations on Reducing the Fall Risk in Older Adults

Figure 6. Effect of Different Intervention Frequencies on Reducing the Fall Risk in Older Adults

We thank you for considering this work and look forward to your response.

Sincerely

Round 2

Reviewer 2 Report

Thank you for the improvements that the authors have made to the manuscript but I continue to believe that methodologically it has many shortcomings and does not add anything new to the already published literature. In the present case, in my opinion, the authors should justify the rationale for this revision in a more concrete way because there are more complete recent reviews.

Author Response

Thank you for your suggestions.In response to the problem you mentioned, we analyzed it, and it should be caused by the following reasons:

(1) The difference in the research objects selected in the study. Our study will not be suitable for the elderly who participate in sports, such as the elderly with dementia, hemiplegia, disability, fractures, etc., so a certain number of articles had been excluded.

(2) The difference in the outcome indicators selected in the study. A large number of existing studies have selected the number of falls and the fall rate as the outcome indicators. However, this indicator has a certain error. A large number of falls without serious consequences are forgotten by the elderly, which greatly reduces the number of reported falls, and some elderly people participate in sports to improve their individual ability and reduce the risk of injury after a fall. A certain number of falls was underreported. The outcome indicators selected in our study are: PPA test score, SOT test score, CS-30 test score, Tetra fall index, Morse fall risk factor assessment and fall risk score. The above indicators are objective indicators for predicting falls, and are not affected by underreporting by the elderly, and the results obtained are more accurate.

Last time you mentioned several recent studies in your comments, and we are very grateful for the articles you mentioned. We also made the following explanation:

The first study is: Comparative effectiveness of exercise interventions for preventing falls in older adults: A secondary analysis of a systematic review with network meta-analysis. This analysis identified components of effective fall prevention exercise. The results can inform evidence-informed exercise recommendations and be used to design effective programs. This research is based on elderly people participating in physical training to reduce the risk of falls, such as: anticipatory control, dynamic stability, functional stability limits, reactive control and flexibility. However, our research topic is about sports. Sports not only include physical training, but also include Tai Chi, Baduan Jin, yoga, Muay Thai dance and other items. However, our research includes as many sports as possible.

The title of the second study is: Evidence on physical activity and falls prevention for people aged 65+ years: systematic review to inform the WHO guidelines on physical activity and sedentary behaviour. Although multiple types of exercise are mentioned in this study He Taijiquan is an evaluation of the effect of reducing the risk of falls in the elderly, but it does not mention the effect of the length of exercise intervention and the frequency of exercise on the effect of reducing the risk of falls. In addition, the outcome indicator selected in the study is the rate of falls. The indicator of fall rate is highly subjective. Many falls that did not cause serious injury are often missed by the elderly because the reported fall rate will be far lower than the actual situation. However, our research selected more scientific outcome indicators, such as: PPA test score, Sensory Organization Test (SOT) test score, 30-second chair stand (CS-30) test score. These indicators have been proven effective.

The title of the third study is: Effectiveness of multifactorial interventions in preventing falls among Older adults in the community: A systematic review and meta-analysis. This study explores the impact of multifactorial interventions on reducing the risk of falls in the elderly, among which Multifactorial interventions included the following components: exercise, education, environmental modification, medication, mobility aids , and vision and psychological management. Movement is just one aspect. Since it is difficult to explain the specific impact of each influencing factor on the results with a multi-factor plan, we would like to explore the impact of exercise on reducing the risk of falls for the elderly, rather than a multi-factor plan.

The title of the fourth study is: Efficacy and Generalizability of Falls Prevention Interventions in Nursing Homes: A Systematic Review and Meta-analysis. The study is more biased towards the care of the elderly, so it includes many dementia elderly people, and the number of included studies is relatively small. many. The focus of our discussion is the elderly group who can exercise autonomously, and the dementia group is excluded. In addition, for the elderly with dementia, there are more safe ways to reduce their risk of falling. Without the caregiver's exercise, it is easy to cause them injury.

The title of the fifth article is: Exercise for preventing falls in older people living in the community: an abridged Cochrane systematic review. It has similar results to the second study, and the outcome indicator is also the fall rate. As mentioned above, the fall rate is a highly subjective indicator with large errors.

The title of the sixth study is: The Effects of Physical Exercise on Balance and Prevention of Falls in Older People: A Systematic Review and Meta-Analysis. The study is more inclined to explain which dimensions of abilities of the elderly improved by sports, such as : Dynamic balance, static balance, participants' fear of falling, balance confidence, quality of life, and physical performance. However, the study did not point out what sports the elderly participate in in order to reduce the risk of falling in the elderly.

Therefore, we have a paragraph in introduction sections from line 76.

We thank you for considering this work and look forward to your response.

Sincerely,

Reviewer 3 Report

Congratulations. The work has been improved considerably. 

Author Response

Thank you for your valuable suggestions and affirmation